# Applying network analysis to assess the development and sustainability of multi-sector coalitions

Tessa Heeren[1]*, Caitlin Ward[2], Daniel Sewell[2], Sato Ashida[3]

**1** Health Policy Research Program, University of Iowa Public Policy Center, Iowa City, IA, United States of America, **2** Biostatistics, University of Iowa College of Public Health, Iowa City, IA, United States of America, **3** Community and Behavioral Health, University of Iowa College of Public Health, Iowa City, IA, United States of America

* Tessa-heeren@uiowa.edu

## Abstract

**Data Availability Statement:** All relevant data are within the article and its Supporting Information files.

**Funding:** The authors received no specific funding for this work.

### Background

Accountable Communities of Health (ACH) models have been popularized through Center for Medicare and Medicaid Innovation (CMMI) grants, including the State Innovation Model (SIM), to encourage the development of community-based coalitions across medical, public health, and social service delivery systems. These models enhance care coordination for patients and are better equipped to address Social Determinants of Health (SDH) needs.

### Methods

Network data was collected from participating organizations in seven ACH sites established across Iowa. The application of network analysis quantitatively characterized the relational context of the interorganizational, cross-sector networks which are foundational to achieving the ACH goal of systematic, comprehensive care. Our analysis primarily used logistic network regression modeling (LNRM) to identify network structures and characteristics of organizations that facilitate or impede sustainable connections.

### Results

Our findings suggest that the ACH was effective at stimulating sustainable connections across sectors and disparate positions of centrality in the network. Factors associated with sustainable connections between organizations included the strength of relationships and the type of collaboration, namely data sharing and resource sharing. Leadership roles designated by the ACH structure were associated with stimulating connections during the grant, but not with sustainment. Network measures of density and transitivity, which peaked during the grant period (compared to pre- and post-grant networks), further implied possible attrition of the ACH intervention effects without incentive to maintain collaborations.

**Competing interests:** The authors have declared that no competing interests exist.

## Conclusions

Multi-sector care coordination networks were established, but our findings suggest depreciation of ACH intervention momentum and structure without incentive to maintain collaborations beyond the three-year duration of the grant. Sustainability could be bolstered and ACH goals actualized with reliable long-term funding.

## Introduction

Healthcare utilization among populations with unmet social determinant of health (SDH) needs is emergent, expensive, and recurring. Patients with inconsistent employment, income, insurance, childcare, transportation, housing and food insecurity experience barriers accessing and adhering to routine preventative care, and as a result require high-cost and reactionary treatments—a burden for both patients and the healthcare system [1, 2]. In response, healthcare policies and practices have increasingly adopted multi-sector efforts to coordinate care inclusive of SDH needs, such as the Accountable Communities of Health (ACH) model.

The ACH model convenes community stakeholders to develop systematic screening and referral systems to "address a critical gap between clinical care and community services" [3]. These referral networks aim to bridge the traditional divide between clinical care, public health, and social services by engaging partners across sectors to deliver comprehensive services. Prior research underlines the importance of coordinated service networks to effectively deliver care and meet various non-clinical determinants of health [4, 5]. The Center for Medicare and Medicaid Innovation (CMMI) promotes the ACH model as a promising practice, and adoption has been promulgated through CMMI grant programs like the State Innovation Model (SIM). In 2016, Iowa committed a portion of its SIM grant funding to establish seven ACH sites across the state.

These three-year ACH awards in Iowa are examples of "network-building" and "network-altering" interventions because of the aims to 1) introduce key actors through the establishment of a governance structure, 2) initiate ties between unconnected actors within and across different sectors, and 3) strengthen ties between actors to collaborate towards a common goal [6, 7]. Interventions which support the formation of trust have been shown to increase the collective social capital and knowledge transfer, meaning we expect networks to demonstrate measurable increase in density through the formation of connections between actors and we expect the quality of connections between actors to become more trusting, reciprocal and functional, improving the sustainability of network change interventions [7–10].

In this article, we use theoretical approaches of social network analysis (SNA) to quantify and characterize the interorganizational relationships that are foundational to ACH operations. Specifically, the application of network analysis assesses whether such network intervention strategies lead to network outcomes considered desirable for investing in lasting system change [9, 11].

Specifically, our analysis provides insight into connectivity within network structures and identifies factors associated with sustainable connections. Our focus on measuring network connectivity through density and dyadic relationships is based in theory which emphasize robust connection across actors in networks to achieve efficient exchanges (i.e., referral network infrastructure employed by ACH models) by closing structural holes of the pre-intervention networks and increasing functional ties [8, 10]. The discontinuation of grant funding presents a challenge, since ACHs will have to tackle the same obstacles that cultivated siloed efforts pre-intervention, including fee-for-service reimbursement, constraints of funding

stream purposes, differing organizational motives and competition, various specializations and capacity to provide services, as well as hierarchy within organizations, sectors, and disciplinary groups [12–15]. Current literature suggests that sustainable connections rely on goal consensus, shared investment (intangible and material), governance structure, accountability, and mutual trust [16–18].

Our findings respond to a gap in the public health coalition literature, as demonstrated by recommendations in prior publications, which have advised alternative methods for quantifying collaborations and more definitive research to understand factors related to sustainment of coalitions [19, 20]. In addition, Bevc, Retrum and Varda (2015) remarked on the lack of knowledge in the formation and evolution of collaborative networks and advised using data-driven methods to inform network performance (particularly for cross-sector endeavors) [16]. Furthermore, Willis (2013) specifically suggested using network analysis to clarify elements of network performance pertinent to evaluation, including sustainment [11]. Given the increased application of multi-sector interventions to provide care that is comprehensive, efficient, and preventative, refining our understanding of interorganizational networks is essential for the development of grant and policy criteria, delineation of coalition operational requirements, and efficient allocation of funding.

## Methods

Beginning in 2015, the independent evaluation of the Iowa SIM documented qualitative reports from site leadership that relationship-building was a success, and that the complexities inherent to cross-sector care coordination (e.g. selecting standardized screening tools) delayed systematic implementation of referral systems within the grant period [21]. From the qualitative data collected, members of ACH networks reported changes to relationships amongst collaborating organizations which allude to principles common in network theory and interorganizational change literature, noting establishment of new connections, increased trust amongst coalition members, a shared vision, and improved collaboration. For example, coalition members shared; "When the [ACH site] formed and work began, some of the partners were not yet at the table but those present needed to work through siloing and some mistrust. Sometimes partnerships simply had not been thought of in the way proposed by the [ACH site]" and "This has been a huge community project where multiple organizations had to be on board with decisions made and think of not only their agency's goals/wants but also that of the greater good." In response, the evaluation team developed an ACH network survey, which was administered to all seven sites in Iowa in the fall of 2018, about two and a half years through the three-year grant. Representatives from organizations included as formalized members of ACH coalitions comprised the network bounds at each site.

The site-level ACH governance structure prescribes leadership roles, including an *integrator organization* (lead facilitator) and a *steering committee* (decision-making body) [21]. The ACH model recommends appointing public health agencies as the integrator organization, perhaps because community organizing is a tenet of the discipline, and public health agencies are in theory neutral when convening competing organizations (like healthcare facilities and social services) [22, 23]. The seven ACH sites in the state were a mix of single county (urban) and multi-county (rural) sites, with the intention of developing models for future replication for both settings.

## Sample

At five of the seven Iowa ACH sites, the integrator organization was a county government-based public health department whereas the remaining two were healthcare-based public

health departments. Other stakeholders in Iowa's ACHs included local representatives from hospitals, primary care providers, other healthcare providers (e.g., behavioral health, pharmacy, dental), insurers, community action organizations, governmental entities, and social service providers. Surveys were completed by the staff designated as primary contact(s) representing the organization in the ACH network. In cases in which multiple staff from a single organization were involved, survey responses were aggregated at the organization level. These representatives comprised the survey sample, which was largely practitioner perspectives (as opposed to formal leadership e.g., CEOs).

Between June and October 2018, an invitation to participate in an online survey was sent to representatives of all ACH-affiliated organizations (n = 169) using the Qualtrics platform. The University of Iowa Institutional Review Board (IRB) determined that this research does not meet the regulatory definition of human subjects research, thus IRB review was not required.

## Data and measures

A roster of all possible collaborators was provided in the survey to minimize recall error and strengthen the equity of reporting, as each respondent was prompted to consider ties with each eligible organization in the network [24]. Respondents of ACH network survey selected all organizations with which their organization collaborated and characterized their partnerships by collaboration type, relationship strength, and evolution of relationship.

## Collaboration types

Respondents were asked about four types of collaborative activities associated with the ACH function: 1) advisory role ("work together to guide the strategic direction"), 2) care coordination ("send and/or receive client care coordination referrals across organizations"), 3) data sharing ("contribute to or have access to shared database, shared client data or other data such as surveys and focus groups across organizations"), and 4) resource sharing ("contribute/ receive resources through collaboration such as money, training/educational materials, space, staff"). Relationships were differentiated by collaboration type to gain insight on the impact of function on strength of collaboration. For instance, some tasks require mutual trust and investment (e.g. data sharing) while execution of other activities (e.g. co-serving on an advisory board) are less intensive [25].

## Strengths of relationships

Strength of relationship was measured via response items indicating level of trust and duration of relationship. For each collaboration, respondents indicated: "little trust/new relationship," "some trust/developing relationship," or "high trust/strong relationship." In response to initial feedback indicating difficulty gauging relationship strengths based on trust alone, response items were emended to include duration of relationship ("new," "developing," "strong"). Establishing rapport among organizational representatives is essential to carry out complex ACH functions, such as making shared decisions in program development and prioritizing a shared vision over organizational autonomy [25]. In turn, trusting relationships normalize collaboration as a routine practice, thus supporting sustainment of the initiative and receptiveness to additional collaboration [9, 26–28].

## Relationship evolution

To assess the SIM grant's goal to stimulate sustainable relationships, respondents were asked to identify the duration of their network collaborations relative to the SIM grant. Respondents

indicated whether each collaboration "preexisted SIM grant" or was "stimulated by SIM participation," as well as to project whether the collaboration was "expected to continue beyond SIM funding." The inclusion of pre-intervention relational contexts helps isolate the impact of the ACH intervention on the network during and after SIM funding.

### Organizational characteristics

Prior to analysis, characteristics of interest were defined and assigned to each ACH-affiliated organization to indicate leadership roles (e.g., integrator organization, steering committee) and sector affiliation (e.g., healthcare, public health, social services).

### Analysis

Based on the survey data, three sets of networks were constructed, with each ACH-affiliated organization represented as a node, and each reported connection represented as an edge: the first network represents preexisting collaborations, the second represents both preexisting and grant-stimulated collaborations (during grant), and the third represents predictions of sustainable collaboration. Descriptive statistics on the preexisting, during grant, and predicted networks were evaluated by site.

Fig 1 is a visualization of the network outcomes at two contrasting ACH sites, selected because they exemplify the variability in network sustainability across sites. The figure displays preexisting and ACH-stimulated ties present during the grant period and predicted sustained networks.

### Factors associated with stimulating connections

A variant of the logistic network regression model (LNRM) proposed by Almquist and Butts [29] was used to evaluate factors associated with stimulation of connections. Two specific hypotheses were proposed in relation to network changes during the ACH grant based on network theory which suggests highly central actors in networks tend to actively form new connections, particularly with similarly influential actors [4]. First, it was hypothesized that organizations already active in collaboration would be more likely to be involved in new collaborations. So, a dyad-level (i.e., pairwise) covariate, the sum of degree centrality (i.e., the total number of connections involving either of the two organizations of the pair), was included to measure relative intranetwork activity for each pair [30]. Secondly, it was hypothesized that actors with relatively similar positions of influence and prominence in the preexisting network would be more likely to form new edges (or make new connections) when stimulated by the SIM grant. We therefore included as a dyad-level covariate the absolute difference in closeness centrality, thus capturing pairwise similarity in centrality of network position [31, 32]. Closeness centrality was calculated as the sum of the inverse distances and measures how close, in terms of network path lengths, an organization is to all other organizations in its ACH site network; in our application, an organization with higher closeness centrality indicates a higher capacity to connect with each other organization in the network efficiently, a key function for an interorganizational referral network [31, 33].

In addition, concordance of organizational sectors was analyzed by including a dyad-level indicator between two organizations in the same sector. To investigate the effect of the ACH governance structure, dyads were examined to note whether the pair of nodes included the integrator organization and/or steering committee members. Lastly, a site-specific intercept was included in the model to account for differences in the number of edges produced by each of the seven sites.

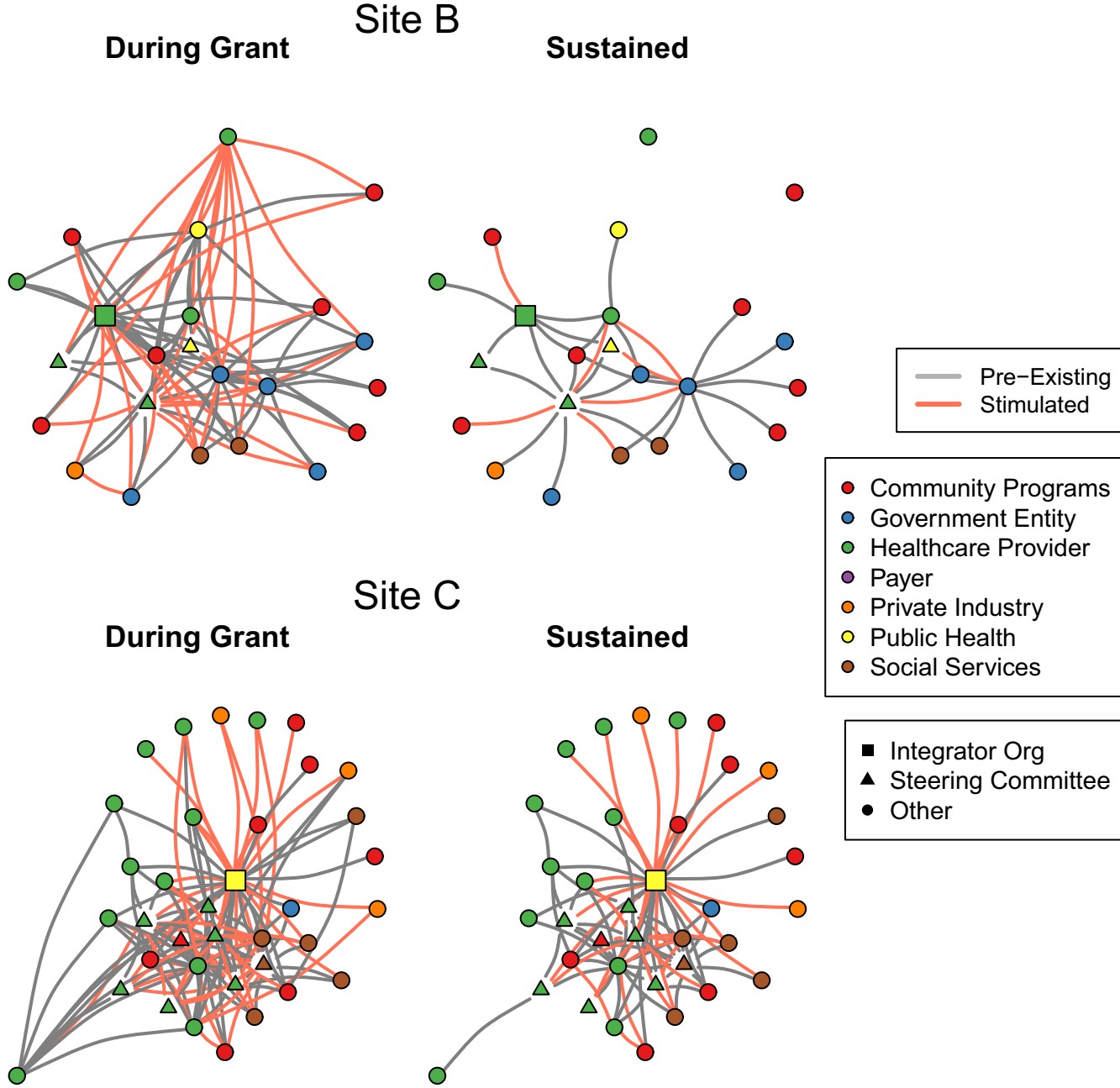

**Fig 1. Visualizations of network summaries of selected ACH sites in Iowa.**

### Factors for sustaining connections

To determine the factors that led to the predicted sustainment or dissolution of existing edges, a sustaining factors LNRM was fit to the data. This approach considered all current edges as the binary outcome (sustained or dissolved). The sum of the organizations' degree and the absolute difference in closeness centrality were included in the model, now based on the active grant period network structure (e.g., preexisting and stimulated network connections). Sector concordance, lead organization, steering committee, and site-specific intercepts were all

included in this model in the same form as in the stimulated LNRM. An indicator for whether the collaboration was newly stimulated or present in the preexisting network was included to test if SIM-stimulated connections were likely to be predicted to be sustained by reporting respondents. In addition to the measures included in the stimulated connections model, the sustained connections model also included the average reported strengths of relationships between dyads (or in the cases where one organization did not respond, the dyad's single reported strength of relationship), as well as the indicator variables corresponding to respondents' characterization of the collaboration's purpose (i.e., advisory role, care coordination, data sharing, and/or resource sharing).

### Collaboration types and stimulated relationships

To investigate whether SIM-stimulated edges tended to favor a particular type of collaboration, chi-square tests were performed to compare the proportion of stimulated edges involving each of the four collaboration types.

### Data missingness

Not all organizations responded to the ACH network survey, thereby providing only partial views of the networks. In cases where one organization claimed a connection with a second organization which was not reciprocated, the pair's connection was assumed, as false negatives appear much more frequently in network data than do false positives [34]. No response from either organization in a pair qualified as a missing dyad. We implemented exponential random graph imputation methods to handle missingness [35, 36].

## Results

### Network changes over time

Table 1 summarizes key network descriptive statistics for each site for preexisting, active grant period, and predicted networks. Specifically, we computed network density (i.e. the ratio of actual to potential edges) and transitivity (i.e. the inclination for two entities with a common connection to form a triad with a third entity, or the often observed tendency to close structural holes) [37]. For all sites, higher density and transitivity measures were observed in the active grant period network compared to preexisting and predicted networks.

### Significant factors in stimulation of new collaborations

Results for the variables of interest from the LNRM predicting stimulated edges are summarized in Table 2, with statistically significant variables bolded. Effects are reported in terms of the odds ratios.

As indicated by the sum of node degree variable, organizations already active in collaboration were more likely to become involved in new collaborations, with additional preexisting connections increasing the probability of stimulation (Odds ratio (OR) = 1.16, 95% Confidence Interval (CI) = [1.10, 1.22]). Contrary to the hypothesis that organizations with more similar closeness centrality in the preexisting network were more likely to form stimulated edges with one another, there was a significant positive effect of the absolute difference in closeness centrality between the organizations (OR = 1.09, 95% CI = [1.03, 1.15]). This statistically significant positive effect implies that organizations with more dissimilar centralities were more likely to form connections, such as a stimulated connection between a central hub and a peripheral organization. In addition, there was not statistically significant evidence of concordance across sectors (OR = 1.3, 95% CI = [0.77, 2.07]).

**Table 1. Iowa ACH site characteristics and network summaries.**

|  | Site A | Site B* | Site C | Site D | Site E* | Site F | Site G |
|---|---|---|---|---|---|---|---|
| *Site characteristics* |  |  |  |  |  |  |  |
| Number of nodes | 28 | 23 | 36 | 11 | 29 | 16 | 26 |
| Response rate | 53.6% | 43.5% | 38.9% | 45.5% | 37.9% | 43.8% | 57.7% |
| Total possible edges | 378 | 253 | 630 | 55 | 406 | 120 | 325 |
| Edges imputed | 20.6% | 27.3% | 36.7% | 27.3% | 37.7% | 30.0% | 16.9% |
| Preexisting edges | 49 | 56 | 81 | 10 | 54 | 41 | 55 |
| Stimulated edges | 29 | 33 | 53 | 4 | 3 | 6 | 19 |
| Likely sustained edges | 55 | 32 | 91 | 10 | 42 | 32 | 49 |
| Steering committee members | 6 | 4 | 9 | 2 | 4 | 2 | 10 |
| *Network summary measures* |  |  |  |  |  |  |  |
| *Density* |  |  |  |  |  |  |  |
| Preexisting | 0.155 | 0.319 | 0.168 | 0.209 | 0.201 | 0.458 | 0.209 |
| Grant period | 0.232 | 0.454 | 0.257 | 0.281 | 0.208 | 0.529 | 0.268 |
| Likely sustained | 0.168 | 0.127 | 0.157 | 0.207 | 0.164 | 0.393 | 0.161 |
| *Transitivity* |  |  |  |  |  |  |  |
| Preexisting | 0.341 | 0.452 | 0.354 | 0.455 | 0.387 | 0.548 | 0.321 |
| Grant period | 0.371 | 0.563 | 0.440 | 0.462 | 0.410 | 0.604 | 0.398 |
| Likely sustained | 0.337 | 0.246 | 0.317 | 0.321 | 0.386 | 0.489 | 0.250 |

* Integrator organization was healthcare-based public health department (as opposed to county government public health)

Note: Summaries are for the imputed networks, presented as summary

**Table 2. Network variables associated with stimulated and likely sustained edges in Iowa ACH sites.**

|  | Stimulated | | | Likely Sustained | | |
|---|---|---|---|---|---|---|
| Variable | Odds Ratio (OR) | 95% confidence Interval (CI) for OR | P-value | Odds Ratio (OR) | 95% confidence Interval (CI) for OR | P-value |
| *Autoregressive measures* |  |  |  |  |  |  |
| Sum of node degree | **1.157** | **(1.097, 1.221)** | **<0.01** | **1.062** | **(1.010, 1.116)** | **0.0192** |
| Similarity = abs diff in closeness cent | **1.092** | **(1.034, 1.154)** | **<0.01** | 1.054 | (0.954, 1.165) | 0.3038 |
| Same sector indicator | 1.259 | (0.767, 2.065) | 0.3625 | 1.440 | (0.873, 2.377) | 0.1531 |
| *ACH structure* |  |  |  |  |  |  |
| Integrator org | **2.715** | **(1.248, 5.909)** | **0.0118** | 1.217 | (0.557, 2.658) | 0.6222 |
| Both members of steering committee | **13.860** | **(6.305, 30.469)** | **<0.01** | 0.887 | (0.347, 2.269) | 0.8023 |
| One member of steering committee | **3.226** | **(1.999, 5.206)** | **<0.01** | 1.230 | (0.688, 2.200) | 0.4852 |
| Average strength of relationship |  |  |  | **1.881** | **(1.377, 2.569)** | **<0.01** |
| Stimulated by SIM |  |  |  | 0.695 | (0.411, 1.173) | 0.1729 |
| *Collaboration types* |  |  |  |  |  |  |
| Advisory |  |  |  | 1.534 | (0.859, 2.739) | 0.1486 |
| Care coordination |  |  |  | 1.264 | (0.763, 2.095) | 0.3630 |
| Data sharing |  |  |  | **2.129** | **(1.265, 3.585)** | **<0.01** |
| Resource sharing |  |  |  | **2.164** | **(1.321, 3.546)** | **<0.01** |

Note: Odds ratio estimates, CIs, and p-values from the Stimulated and likely Sustained Logistic Network Regression Models (LNRM)

Introducing key actors to the network through formal leadership roles, such as the integrator organization was an important factor in the production of stimulated edges (OR = 2.7, 95% CI = [1.25, 5.91]). Similarly, steering committee organizations made more connections with other organizations, where having one or both organizations in a pair greatly increased the odds of a stimulated edge (OR = 3.23, 95% CI = [2.00, 5.21] and OR = 13.86, 95% CI = [6.31, 30.47], respectively).

## Significant factors in sustainment of collaborations

Table 2 displays the results of the LNRM analyzing factors leading to predicted sustainment of a preexisting or stimulated edge.

There was a marginally significant effect of the sum of the degree between the two organizations (OR = 1.06, 95% CI = [1.01, 1.12]). The effect of the difference in closeness centrality was not statistically significant. The involvement of the integrator organization and the steering committee did not significantly affect predicted sustainment of collaborations. There was also no significant difference in predicted sustainment between preexisting and newly formed edges.

Higher levels of trust were significantly associated with increased probability of sustaining a connection (OR = 1.88, 95% CI = [1.38, 2.57]). Furthermore, the likelihood of an edge's sustainment was increased if the relationship involved data sharing (OR = 2.13, 95% CI = [1.27, 3.59]) or sharing resources (OR = 2.16, 95% CI = [1.32, 3.55]).

Fig 2 depicts the predicted probability of an active grant period edge's future sustainment, based on the average trust of two organizations for each site. The graph illustrates the increase in the probability of a sustained relationship as the trust between two organizations increases.

## Chi-square test results

The results of the chi-square tests comparing stimulated and preexisting edges according to collaboration type indicate that a significantly higher proportion of preexisting edges report data sharing and resource sharing. Preexisting edges are almost twice as likely (OR = 1.90, 95% CI = [1.42, 2.53]) to share data and 26% more likely to share resources (OR = 1.26, 95% CI = [1.06, 1.49]). The difference between the proportion of preexisting and stimulated edges sharing an advisory role (OR = 1.13, 95% CI = [0.85, 1.49]) or participating in care coordination (OR = 0.95, 95% CI = [0.82, 1.11]) was not statistically significant.

## Site characteristics

While comparisons between sites was not the primary objective, some findings add to our understanding of the processes under study. Of these site-site comparisons, 11 results were significant, and two sites surfaced as less successful at stimulating and building sustainable relationships. Compared to the other ACH sites, Site E had significantly lower odds of stimulated edges, whereas Site B had significantly lower odds of sustaining edges. Sites A and D had the highest probabilities of sustaining connections. The characteristics of the lead organizations at sites may be a factor, as Sites B and E were the only sites of the seven led by healthcare entities (the other five were public health-led sites).

## Discussion and conclusion

This extensive network analysis of the seven ACH sites in Iowa provides insight about key ACH relationship factors that may influence stimulation and sustainment of organizational collaborations. Our research questions examined the application of three tenets of network

## Predicted Probability of a Sustained Edge
## by Average Relationship Strength and Site

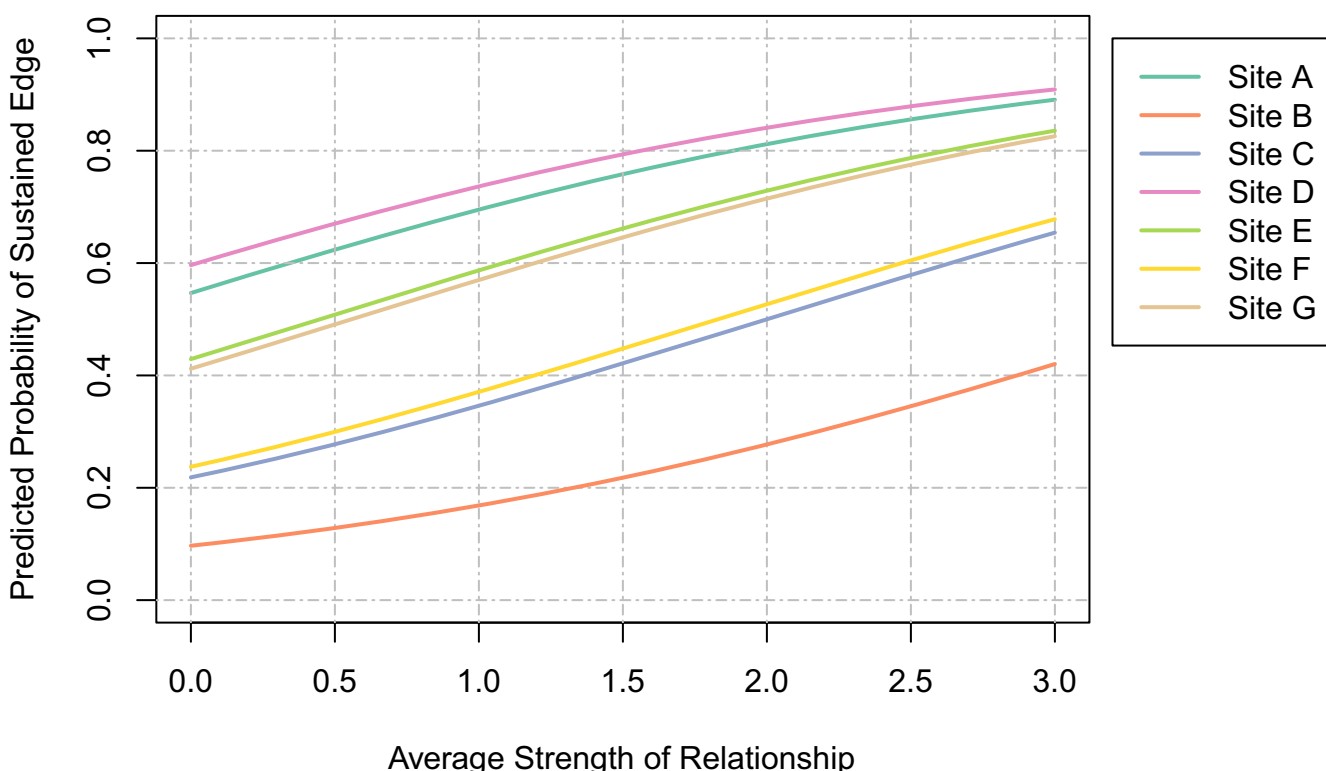

**Fig 2. Iowa ACH site-level relationships between predicted probability of edge sustainment and average self-rated strength of relationship.**

interventions: 1) the introduction of key actors in the network, 2) the formation of ties between unconnected actors, and 3) the sustainability of ties. The results elucidated that the introduction of the integrator organizations likely led to increased connectivity among organizations, and several organizational, as well as relationship characteristics, were associated with the development and predicted sustainment of cross-organizational collaborations. Three prominent themes emerged from the network data: 1) ACH membership and structure, 2) increased connectivity, and 3) sustainment.

### ACH membership and structure

The ACH governance structure had a statistically significant role in relationship formation. Dyads that included the integrator organization or at least one steering committee member were significantly more likely to report a newly stimulated connection compared to dyads without a leadership player. This result implies that the ACH-prescribed governance structure succeeded in introducing key players to the network, as formalized leaders were actively involved in stimulating network connections. Conversely, leadership designations as the integrator organization and steering committee and use of formal agreements were not significant predictors of collaborations likely to be sustained, suggesting that the structure introduced by the ACH model may be susceptible to post-funding attrition. In addition, active community collaborators prior to funding (e.g., dyads with many preexisting network connections) played significant roles in network connection stimulation.

At the ACH level, two networks were relatively unsuccessful at stimulating and building sustainable relationships (Sites E and B, respectively). The sectoral affiliation of the integrator organization could potentially explain these differences, as these two sites were led by health-care-based public health departments, while government public health agencies led the other five ACH sites. The healthcare-based integrator organizations may have struggled to gain buy-in from competing healthcare providers in the community, as was suggested in qualitative results in SIM evaluation reports [21].

## Increased connectivity

Across the seven Iowa ACH sites, a total of 147 collaborations (18% of 804 total potential connections) were identified as "stimulated by the SIM," and concordance results suggest that cross-sector collaborations may be just as likely to be stimulated as same-sector collaborations. In addition, the significance of the absolute difference in closeness centrality indicates that organizations of different levels of centrality were more likely to form collaborations. These findings suggest that the Iowa SIM activities succeeded as a network intervention by stimulating collaborations between central and peripheral organizations in the network and across sectors to build a diverse coalition capable of innovation.

The ACH model requires multilayered collaborations to succeed in coordinating care across sectors. Of the various functions of the ACH, the tasks that required intense collaboration—namely data and resource sharing—were more likely to involve network actors with pre-existing connections, which were also more likely to be sustained. These results are aligned with literature suggesting that collaboratively intensive activities (e.g. cooperatively selecting data collection tools) are more time-consuming to establish, but an initial investment can enhance continued momentum [38]. While care coordination is the ultimate goal, this activity was not significant to stimulating or sustaining connections. This could be due to the execution of care coordination by a subset of organizations, or because the time spent establishing foundations for care coordination (e.g., developing data sharing infrastructure) delayed care coordination network development.

## Sustainability

Several results from this analysis underscore the importance of consistent and stable relationships for successful ACH-driven collaboration, which is in alignment with previous literature [39–41]. The ACH appeared to have successfully facilitated sustainable collaborations, since edges stimulated by the ACH were not statistically significantly different from preexisting edges. While ACH-stimulated sustainable relationships were independent of preexisting relationships, the quantity of established relationships in general had a role in determining sustainable connections. The significance of sum degree in both models suggests a feedback loop: the more an organization engages in collaboration, the more it is likely to form and sustain collaborations in the future. A high level of previous engagement may indicate an organizational or local culture of collaboration.

Notably, strength of relationship factored strongly into whether organizations anticipated sustaining a collaboration. This result is in keeping with literature demonstrating that high trust sustains relationships critical to effective care coordination [26–28].

The network building intervention may degenerate without funding, as indicated by the lessened role of ACH leadership organizations in sustainable connections. Additionally, the density and transitivity of all ACH networks (indicators of network efficiency and connectivity) peaked during the grant period, with measures dropping to pre-intervention levels in the predicted sustaining networks.

## Limitations

Because the ACH network survey was added to the preestablished evaluation plan midway through the formal evaluation period, organizations were not required to comply. While analytic methods were used to minimize the impact of incomplete participation, reported connections may not reflect all existing connections. A major limitation of this study is the use of self-reported and cross-sectional data to evaluate past, present, and potential future connections; however, a small simulation study suggested that our results were highly robust to recall error of pre-existing connections between organizations. Another limitation due to using cross-sectional data is that we were unable to confirm that the self-reported sustained connections were in fact sustained, and in this sense it may be better to interpret our results in terms of intent to sustain.

## Implications

As these results demonstrate, selectively awarded, time-limited grants cannot initiate and sustain robust cross-sector collaboration capable of transforming the healthcare system and facilitating SDH care; however, as also reinforced here, local public health agencies are capable of convening efficient, cost-curving care coordination networks. Reliable funding that supports relationship-building over time could allow for effective preventative population health interventions [39–41]. Although reform from prevalent fee-for-service payment structures remains a challenge in Iowa and nationally [21, 40, 42], funding to support ACH-like care coordination could be sourced from value-based (as opposed to volume-based) healthcare reimbursements.

Within the current healthcare landscape of grant-funded initiatives, strategies to establish and strengthen relationships and trust should include explicit expectations in outcomes and measures to ensure accountability. Without a government mandate to institute payment reform or resource allocation to support public health agencies, the void in network leadership removes the compulsion and reduces accountability, which can lead to network dissolution [7, 22, 43]. Along with funding a designated network manager, all participating stakeholders need financial viability to compensate for immediate losses and increased expenditures (e.g. infrastructure, staff time) [39]. Opportunities for future research include rigorous cost-benefit analysis of fully functioning ACH systems to develop a business case for funding.

Network analysis characterizes relationships, which could be used to create measures to monitor network formation and development and promote accountability. For example, network analysis could inform recruitment for ACH membership. By identifying prominent community organizations and introducing peripheral organizations through the ACH, community connections across dissimilar organizations could be purposefully stimulated. Finally, facilitated reflection on network data can strengthen networks by prompting members to identify as part of a cooperative network, explore roles and goals, and acknowledge conflict [44].

## Supporting information

**S1 File. Data for Fig 2.** Authors' LNRM results of ACH network survey data. (XLSX)

**S2 File. ACH survey instrument.** Survey distributed via email to ACH survey sample. (PDF)

## Author Contributions

**Conceptualization:** Tessa Heeren, Daniel Sewell, Sato Ashida.

**Data curation:** Tessa Heeren.

**Formal analysis:** Caitlin Ward, Daniel Sewell.

**Investigation:** Tessa Heeren.

**Methodology:** Tessa Heeren, Caitlin Ward, Daniel Sewell, Sato Ashida.

**Project administration:** Tessa Heeren.

**Supervision:** Daniel Sewell, Sato Ashida.

**Validation:** Caitlin Ward, Daniel Sewell, Sato Ashida.

**Visualization:** Caitlin Ward, Daniel Sewell.

**Writing – original draft:** Tessa Heeren, Caitlin Ward, Daniel Sewell, Sato Ashida.

**Writing – review & editing:** Tessa Heeren, Caitlin Ward, Daniel Sewell, Sato Ashida.

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
