## [Decision Letter · Decision Letter 0]

6 Oct 2021

PONE-D-21-13590Applying Network Analysis to Assess the Development and Sustainability of Multi-Sector CoalitionsPLOS ONE

Dear Dr. Heeren,

Thank you for submitting your manuscript to PLOS ONE. After careful consideration, we feel that it has merit but does not fully meet PLOS ONE’s publication criteria as it currently stands. Therefore, we invite you to submit a revised version of the manuscript that addresses the points raised during the review process.

The reviewers have identified several important aspects of your study design that require further clarification and contextualisation. Please attend carefully to the each of the concerns they have raised when preparing your revisions.

We look forward to receiving your revised manuscript.

Kind regards,

Jamie Males

Staff Editor

PLOS ONE

Journal Requirements:

2. Please include additional information regarding the survey or questionnaire used in the study and ensure that you have provided sufficient details that others could replicate the analyses. For instance, if you developed a questionnaire as part of this study and it is not under a copyright more restrictive than CC-BY, please include a copy, in both the original language and English, as Supporting Information. Moreover, please include more details on how the questionnaire was pre-tested, and whether it was validated.

Reviewers' comments:

Reviewer's Responses to Questions

**Comments to the Author**

1. Is the manuscript technically sound, and do the data support the conclusions?

Reviewer #1: Yes

Reviewer #2: Partly

2. Has the statistical analysis been performed appropriately and rigorously? 

Reviewer #1: I Don't Know

Reviewer #2: Yes

3. Have the authors made all data underlying the findings in their manuscript fully available?

Reviewer #1: Yes

Reviewer #2: Yes

4. Is the manuscript presented in an intelligible fashion and written in standard English?

Reviewer #1: Yes

Reviewer #2: Yes

5. Review Comments to the Author

Reviewer #1: I enjoyed reading this paper. It is clearly and succinctly written and reports from an interesting study. I am not equipped to interrogate all of the methods but overall my sense is that the analyses are robust.

The study design matches its aims, and I particularly liked the attempt to measure relationships pre- and post the initiative. The paper reports some important findings, especially re the limitations of such funding approaches when relationship-building requires a longer view.

I think the exclusively quantitative design is a limitation as the tendency is to over-simplify complex social and organisational variables. Some qualitative insights might have added nuance here.

My only suggestion for amending the paper is to elaborate a little on its contribution. The authors talk about ‘refining’ understanding but this could be tied to a prior assessment of limitations/gaps in the literature.

Reviewer #2: This is a very interesting paper, but I would urge you all to go back and think about why you are doing this study, motivate with theory, what are you are capturing in the “networks” and justify your dyadic measures more.

There is no theoretical motivation for this research provided. Secondly, what are the networks you are trying to examine? How is each network bounded? Please explanation what the networks are, at least in theory. And what is the rationale for the dyad level measures? Please explain/give more motivation. Can we assume that most of the organizations have dyadic relationships? What about triads? Or clique structures?

Please explain the creation of the dyad-level covariate which is the absolute difference in closeness centrality, thus capturing pairwise similarity in centrality of

network position. There are many centrality measures. Why is closeness centrality of theoretical or empirical importance herein?

What was the extent of the missing data? Please elaborate on this.

6. PLOS authors have the option to publish the peer review history of their article (what does this mean?). If published, this will include your full peer review and any attached files.

Reviewer #1: No

Reviewer #2: No

---

## [Author Response · Author response to Decision Letter 0]

21 Feb 2022

5. Review Comments to the Author 

Reviewer #1: I enjoyed reading this paper. It is clearly and succinctly written and reports from an interesting study. I am not equipped to interrogate all of the methods but overall my sense is that the analyses are robust. 

The study design matches its aims, and I particularly liked the attempt to measure relationships pre- and post the initiative. The paper reports some important findings, especially re the limitations of such funding approaches when relationship-building requires a longer view. 

I think the exclusively quantitative design is a limitation as the tendency is to over-simplify complex social and organisational variables. Some qualitative insights might have added nuance here. 

Author response: Additional qualitative content added to provide additional context and illustrate motivation for network analysis methodology. 

My only suggestion for amending the paper is to elaborate a little on its contribution. The authors talk about ‘refining’ understanding but this could be tied to a prior assessment of limitations/gaps in the literature. 

Author response: A summary of contributions to documented gaps in literature have been added. 

Reviewer #2: This is a very interesting paper, but I would urge you all to go back and think about why you are doing this study, motivate with theory, what are you are capturing in the “networks” and justify your dyadic measures more. 

There is no theoretical motivation for this research provided. Secondly, what are the networks you are trying to examine? How is each network bounded? Please explanation what the networks are, at least in theory. And what is the rationale for the dyad level measures? Please explain/give more motivation. Can we assume that most of the organizations have dyadic relationships? What about triads? Or clique structures? 

Author response: 

More explicit definition of network boundaries and membership added (formal membership in health coalitions). 

Underlying theory expounded upon (including structural holes, functional ties, and sustaining system change). Also referenced influencing implications from practical applications. 

Theoretical justification for examining dyad-level measures and network density were added (focus on referral network connectivity between organizations and overall were of primary interest related to research question) 

Triads were measured via network transitivity, but wasn’t expounded upon in discussion, as results were not integral to hypothesized outcomes 

Please explain the creation of the dyad-level covariate which is the absolute difference in closeness centrality, thus capturing pairwise similarity in centrality of network position. There are many centrality measures. Why is closeness centrality of theoretical or empirical importance herein? 

Author response: 

Added theoretical content justifying the applied function of closeness centrality in a referral network (e.g. indicates capacity to connect efficiently with each actor in the network) 

What was the extent of the missing data? Please elaborate on this. 

Author response: Table 1 revised to include % of missing edges

---

## [Decision Letter · Decision Letter 1]

5 Jul 2022

PONE-D-21-13590R1Applying Network Analysis to Assess the Development and Sustainability of Multi-Sector CoalitionsPLOS ONE

Dear Dr. Heeren,

Thank you for submitting your manuscript to PLOS ONE. After careful consideration, we feel that it has merit but does not fully meet PLOS ONE’s publication criteria as it currently stands. Therefore, we invite you to submit a revised version of the manuscript that addresses the points raised during the review process.

As you can see, both reviewers suggest relatively minor changes.  Please respond to these comments in your revised manuscript, including the suggestion of adding one more model indicated by Reviewer 4. 

We look forward to receiving your revised manuscript.

Kind regards,

Cynthia Lakon, PhD, MPH

Guest Editor

PLOS ONE

Journal Requirements:

Reviewers' comments:

Reviewer's Responses to Questions

**Comments to the Author**

1. If the authors have adequately addressed your comments raised in a previous round of review and you feel that this manuscript is now acceptable for publication, you may indicate that here to bypass the “Comments to the Author” section, enter your conflict of interest statement in the “Confidential to Editor” section, and submit your "Accept" recommendation.

Reviewer #3: All comments have been addressed

Reviewer #4: (No Response)

2. Is the manuscript technically sound, and do the data support the conclusions?

Reviewer #3: Yes

Reviewer #4: Yes

3. Has the statistical analysis been performed appropriately and rigorously? 

Reviewer #3: Yes

Reviewer #4: Yes

4. Have the authors made all data underlying the findings in their manuscript fully available?

Reviewer #3: Yes

Reviewer #4: Yes

5. Is the manuscript presented in an intelligible fashion and written in standard English?

Reviewer #3: Yes

Reviewer #4: Yes

6. Review Comments to the Author

Reviewer #3: This paper reports on an inter-organizational analysis relationship formed within the context of a state-wide granting mechanism in response to field reports on the challenges of creating and sustaining cross-sector collaborations. A few comments to improve the study.

The one glaring issue, which is certainly not amenable is the reliance on one time period to measures pre-SIM, during SIM, and projected sustainment. Although I am not immediately aware of any such studies I am certain there are other studies which have assessed networks at multiple time points at one time. I would encourage the authors to find such studies, reference, and see how they describe this limitation. Moreover, I wonder if the authors can explore any ways in which they can “validate” these reports.

Second, the authors make no mention of who specifically, within each organization, completed the survey. Presumably each organization has multiple hierarchical levels ranging from CEO, Director, program manager, staff, research assistant, etc. Who was responsible for providing this information can have implications for what they can report. Indeed, a CEO is likely to have quite a different perception of the working relationships than a program manager. This should be analyzed as it might be that the integrator organization consisting of health service agencies may have delegated reporting (survey completion) differently than other organizations.

Line 107: A roster of all possible collaborators was roved to minimize recall error and strengthen the equity of reporting, a

“roved”? -> “provided

Line 120: For instance, some tasks require mutual trust and investment (e.g. data sharing) while execution of other activities (e.g. co-serving on an advisory board) are less intensive.(18)

Should this be “ … less interactive.”?

Line 147: you should cite Freeman, 1979 on closeness centrality and report which variant you use.

Lines 196-197: as per comment 2 above, was there only one individual reporting for each organization?

Lines 220-223: “Contrary to the hypothesis that organizations with more similar centrality in the preexisting network were more likely to form stimulated edges with one another, there was a significant positive effect of the absolute difference in closeness centrality between the organizations (OR = 1.10, 95% CI = [1.04, 1.16]).”

Lines 224-225: “Meaning, organizations with dissimilar centralities (i.e. pairs comprised of one core and one periphery organization) were more likely to form connections.”

Make a visual inspection to be certain these are new core-periphery ties, just because they closeness centrality differences are associated with new ties does not necessarily imply that.

But I would think that organizations with high and similar closeness centrality scores would already be connected. Is this controlled for? At the very least they would have a shared partner thus more likely to become connected.

It might be worth noting if these 7 sites were the 7 largest municipalities in Iowa (or not).

Lines 124-125: “For each collaboration, respondents indicated: “little trust/new relationship,” “some trust/developing relationship,” or “high trust/strong relationship.”

You might note as a limitation that trust and duration of relationship are not synonymous.

Reviewer #4: One suggestion I had is that in the “likely sustained” model, you might add an additional model that only includes the variables in the “stimulated” model. This would allow you to assess the degree to which these added variables (including strength of ties) might moderate the effects of your ACH structure variables. This potentially could provide additional insights.

7. PLOS authors have the option to publish the peer review history of their article (what does this mean?). If published, this will include your full peer review and any attached files.

Reviewer #3: **Yes: **Thomas W Valente

Reviewer #4: No

---

## [Author Response · Author response to Decision Letter 1]

7 Sep 2022

6. Review Comments to the Author

Reviewer #3: This paper reports on an inter-organizational analysis relationship formed within the context of a state-wide granting mechanism in response to field reports on the challenges of creating and sustaining cross-sector collaborations. A few comments to improve the study.

☒The one glaring issue, which is certainly not amenable is the reliance on one time period to measures pre-SIM, during SIM, and projected sustainment. Although I am not immediately aware of any such studies I am certain there are other studies which have assessed networks at multiple time points at one time. I would encourage the authors to find such studies, reference, and see how they describe this limitation. Moreover, I wonder if the authors can explore any ways in which they can “validate” these reports.

The reviewer is correct, in both that this is a major limitation and one which due to study constraints we were unable to avoid. If the reviewer can point us to specific papers we would take them into consideration; however, we are unaware of any papers that investigate the potential ramifications of this study design. The rationale behind perceptions of predicted sustainment as a valid assessment is based in the Theory of Planned Behavior, which states that behavioral intention (in this case, sustaining a relationship) is associated with behavioral achievement (Azjen, 1991). Several prior studies investigating network change over time have called for additional research with longitudinal analysis including variables like trust, stage of relationship, collaborative task, and network size (Bryson, 2015; Provan 2012; Varda, 2008) The Provan 2012 study used longitudinal data, but noted this did not mitigate all the complexities of interpreting network change over time, “First, despite use of longitudinal data, we were unable to develop a full understanding of how network ties evolved. For example, more in-depth qualitative data might have enabled us to examine the impact of [Value Option, a behavioral health network] VO’s contract rules on the evolution of the network as a whole… we did not know how many relationships were already in place when the VO system first was established. Thus, while the VO system was new, many relationships among providers may have predated VO.”(p.373). 

 As a small step in this direction, we ran a simulation study, replicating the analysis for stimulated connections. In this study, we simulated 100 datasets that, in order to emulate recall error, we randomly deleted 10% of the pre-existing edges. We then, for each simulated dataset, computed the estimated odds ratio and confidence intervals. We found remarkably little variation in the estimates, as can be seen in the figure below, which shows the boxplots of the 100 simulations for the lower bound and upper bound of the confidence intervals as well as for the point estimate of the odds ratio. The original CI bounds and point estimates are superimposed in red asterisks. It is clear that even if there is poor recall in pre-existing connections between organizations, we are still able to obtain accurate inference.

Regarding the analysis of sustained edges, since we were unable to confirm whether or not a connection was in fact sustained, it might be better to interpret our results in terms of intent to sustain. We have changed the Limitations Section to reflect all these points. 

☒Second, the authors make no mention of who specifically, within each organization, completed the survey. Presumably each organization has multiple hierarchical levels ranging from CEO, Director, program manager, staff, research assistant, etc. Who was responsible for providing this information can have implications for what they can report. Indeed, a CEO is likely to have quite a different perception of the working relationships than a program manager. This should be analyzed as it might be that the integrator organization consisting of health service agencies may have delegated reporting (survey completion) differently than other organizations.

Surveys were completed by the staff designated as primary contact(s) representing the organization in the ACH network. In cases in which multiple staff from a single organization were involved, survey responses were aggregated at the organization level. The survey sample was largely comprised of practitioner roles (as opposed to formal leadership roles in each organization), for example, primary care providers, local public health department staff, hospital clinic managers, behavioral health providers, school nurses, law enforcement staff (e.g. sheriff), community members, transportation providers, faith-based organization staff, pharmacists, health coaches, care coordinators, case managers, benefits program administrators (e.g. WIC), non-profit staff, local government representatives (e.g. city managers), community group leaders (e.g. local NAACP chapters), and program managers.

This phrase was added to line 136

Other stakeholders in Iowa’s ACHs included local representatives from hospitals, primary care providers, other healthcare providers (e.g., behavioral health, pharmacy, dental), insurers, community action organizations, governmental entities, and social service providers. Surveys were completed by the staff designated as primary contact(s) representing the organization in the ACH network. In cases in which multiple staff from a single organization were involved, survey responses were aggregated at the organization level. These representatives comprised the survey sample, which was largely practitioner perspectives (as opposed to formal leadership e.g., CEOs)

☒Line 139: A roster of all possible collaborators was roved to minimize recall error and strengthen the equity of reporting, a

“roved”? -> “provided

Line 139, previously corrected in November 2021 revision response

☒Line 153: For instance, some tasks require mutual trust and investment (e.g. data sharing) while execution of other activities (e.g. co-serving on an advisory board) are less intensive.(18)

Should this be “ … less interactive.”?

Line 153, changed to interactive

☒Line 179: you should cite Freeman, 1979 on closeness centrality and report which variant you use.

Freeman 1978 citation added to line 201, in addition to the phrase below, added to line 202 

Closeness centrality was calculated as the sum of the inverse distances and measures how close..

This method of calculating closeness centrality is sometimes called “harmonic centrality,” a type of relative measure (as described in Freeman 1979). The specific variant employed was not described in the Freeman paper.

☒Lines233-234: as per comment 2 above, was there only one individual reporting for each organization?

No, see response to second comment RE sample

Lines262-263: “Contrary to the hypothesis that organizations with more similar centrality in the preexisting network were more likely to form stimulated edges with one another, there was a significant positive effect of the absolute difference in closeness centrality between the organizations (OR = 1.10, 95% CI = [1.04, 1.16]).”

☒Lines 264-265: “Meaning, organizations with dissimilar centralities (i.e. pairs comprised of one core and one periphery organization) were more likely to form connections.”

Make a visual inspection to be certain these are new core-periphery ties, just because they closeness centrality differences are associated with new ties does not necessarily imply that.

We acknowledge that our original language was too strong regarding stimulated edges between core and peripheral organizations. We have changed the text now to read, 

“This statistically significant positive effect implies that organizations with more dissimilar centralities were more likely to form connections, such as a stimulated connection between a central hub and a peripheral organization.”

☒But I would think that organizations with high and similar closeness centrality scores would already be connected. Is this controlled for? At the very least they would have a shared partner thus more likely to become connected.

The analysis for stimulated connections was implicitly conditioning on the non-existence of the edge; that is, we are considering only dyads without a connecting edge, and seeing which factors predict a new connection being formed (if the reviewer is familiar with STERGM, this is akin to the “relational formation” half of the separable likelihood). In the analysis for sustained connections, we are implicitly conditioning on the edge existing (again, akin to the “relational dissolution” half of the STERGM likelihood), and we are also including as a covariate whether or not the edge was originally connected or newly stimulated. 

☒It might be worth noting if these 7 sites were the 7 largest municipalities in Iowa (or not).

This phrase has been added to line 127 – 

The seven ACH sites in the state were a mix of single county (urban) and multi-county (rural) sites, with the intention of developing models for future replication for both settings. 

☒Lines156-157: “For each collaboration, respondents indicated: “little trust/new relationship,” “some trust/developing relationship,” or “high trust/strong relationship.”

You might note as a limitation that trust and duration of relationship are not synonymous.

In pilot survey testing, items were specific only to level of trust. Test respondents reported they were hesitant to rate collaborative relationships as lacking trust. While trust and duration of relationship are not synonymous, alternative language describing the stage of the relationship (i.e., new, developing, strong) was included to mitigate social desirability bias. 

☒Reviewer #4: One suggestion I had is that in the “likely sustained” model, you might add an additional model that only includes the variables in the “stimulated” model. This would allow you to assess the degree to which these added variables (including strength of ties) might moderate the effects of your ACH structure variables. This potentially could provide additional insights.

We appreciate the suggestion from Reviewer 4. We have rerun the analysis for the sustained connections model, and our results are consistent with previously reported findings (see table below). Analysis was completed, results consistent with original findings, suggesting that there are not moderation effects. Because these results are consistent, i.e., there does not appear to be moderation effects to report on, for the sake of brevity we have opted for omitting this additional analysis in the manuscript.

 Stimulated Likely Sustained Likely Sustained (reviewer #4) 

Variable Odds Ratio (OR) 95% confidence Interval (CI) for OR P-value Odds Ratio (OR) 95% confidence Interval (CI) for OR P-value Odds Ratio (OR) 95% confidence Interval (CI) for OR P-value 

Autoregressive measures 

Sum of node degree 1.157 (1.097, 1.221) <0.01 1.062 (1.010, 1.116) 0.0192 1.097 (1.049, 1.147) <0.01 

Similarity = abs diff in closeness cent 1.092 (1.034, 1.154) <0.01 1.054 (0.954, 1.165) 0.3038 1.041 (0.951, 1.140) 0.3843 

Same sector indicator 1.259 (0.767, 2.065) 0.3625 1.440 (0.873, 2.377) 0.1531 1.744 (1.104, 2.754) 0.0171 

ACH structure 

Integrator org 2.715 (1.248, 5.909) 0.0118 1.217 (0.557, 2.658) 0.6222 1.626 (0.805, 3.286) 0.1756 

Both members of steering committee 13.860 (6.305, 30.469) <0.01 0.887 (0.347, 2.269) 0.8023 0.824 (0.385, 1.765) 0.6182 

One member of steering committee 3.226 (1.999, 5.206) <0.01 1.230 (0.688, 2.200) 0.4852 0.886 (0.527, 1.490) 0.6480 

Average strength of relationship 1.881 (1.377, 2.569) <0.01 

Stimulated by SIM 0.695 (0.411, 1.173) 0.1729 

Collaboration types 

Advisory 1.534 (0.859, 2.739) 0.1486 

Care coordination 1.264 (0.763, 2.095) 0.3630 

Data sharing 2.129 (1.265, 3.585) <0.01 

Resource sharing 2.164 (1.321, 3.546) <0.01 

---

## [Editor Report · Decision Letter 2]

29 Sep 2022

Applying Network Analysis to Assess the Development and Sustainability of Multi-Sector Coalitions

PONE-D-21-13590R2

Dear Dr. Heeren,

We’re pleased to inform you that your manuscript has been judged scientifically suitable for publication and will be formally accepted for publication once it meets all outstanding technical requirements.

Kind regards,

Cynthia Lakon, PhD, MPH

Guest Editor

PLOS ONE

Additional Editor Comments (optional):

Thank you for your revisions, they have addressed the remaining comments.
---

## [Editor Report · Acceptance letter]

7 Oct 2022

PONE-D-21-13590R2 

Applying Network Analysis to Assess the Development and Sustainability of Multi-Sector Coalitions 

Dear Dr. Heeren:

I'm pleased to inform you that your manuscript has been deemed suitable for publication in PLOS ONE. Congratulations! Your manuscript is now with our production department. 

Kind regards, 

on behalf of

Dr. Cynthia Lakon 

Guest Editor

PLOS ONE